# The Mechanisms of Social Immunity Against Fungal Infections in Eusocial Insects

**DOI:** 10.3390/toxins11050244

**Published:** 2019-04-29

**Authors:** Long Liu, Xing-Ying Zhao, Qing-Bo Tang, Chao-Liang Lei, Qiu-Ying Huang

**Affiliations:** 1Hubei Insect Resources Utilization and Sustainable Pest Management Key Laboratory, Huazhong Agricultural University, Wuhan 430070, China; lliu1988@henau.edu.cn (L.L.); zxy0818@webmail.hzau.edu.cn (X.-Y.Z.); ioir@mail.hzau.edu.cn (C.-L.L.); 2Plant Protection College, Henan Agricultural University, Zhengzhou 450002, China; qbtang@henau.edu.cn

**Keywords:** social insects, fungal pathogens, social immunity, behavioral and physiological adaptations, social interaction network

## Abstract

Entomopathogenic fungus as well as their toxins is a natural threat surrounding social insect colonies. To defend against them, social insects have evolved a series of unique disease defenses at the colony level, which consists of behavioral and physiological adaptations. These colony-level defenses can reduce the infection and poisoning risk and improve the survival of societal members, and is known as social immunity. In this review, we discuss how social immunity enables the insect colony to avoid, resist and tolerate fungal pathogens. To understand the molecular basis of social immunity, we highlight several genetic elements and biochemical factors that drive the colony-level defense, which needs further verification. We discuss the chemosensory genes in regulating social behaviors, the antifungal secretions such as some insect venoms in external defense and the immune priming in internal defense. To conclude, we show the possible driving force of the fungal toxins for the evolution of social immunity. Throughout the review, we propose several questions involved in social immunity extended from some phenomena that have been reported. We hope our review about social ‘host–fungal pathogen’ interactions will help us further understand the mechanism of social immunity in eusocial insects.

## 1. Introduction

Social insects such as termites, ants, bees and wasps benefit a lot from sociality compared with solitary insects. They can collectively perform nesting, caring, foraging, and defense, which significantly improves the survival of group members [1,2,3]. However, living in group may contribute to the risk of epidemic outbreak. This is because: (a) some social insects live in microbe-rich environments [4]; (b) insect colonies are crowded with closely related members [5]; (c) group members show a high frequency of social contacts [1]. In fact, insect colonies rarely die of diseases. To defend against pathogens that cause disease, social insects have evolved a series of sophisticated disease defenses at colony and individual levels [1,2,3]. In particularly, the colony-level disease defense includes novel behavioral and physiological adaptations termed as social immunity [1,2,3,6,7]. Moreover, the organization of insect societies mediated by social communications and behaviors also contribute to social immunity [6,8,9]. Over the last decades, the functional mechanism of social immunity in insects was reported in a growing number of studies.

Fungal pathogens such as *Metarhizium* and *Beauveria* have been important material for studying social immunity in insects [10,11,12,13,14,15,16]. They can produce a large number of infectious spores or conidia to infect insects, which are widely distributed around insect colonies [3,17]. Once attaching to insect cuticles, fungal pathogens infect the insect hosts via invading body cavity, spreading in vivo, damaging host cells and finally killing the host [18,19,20]. Simultaneously, fungal pathogens produce some toxins to facilitate their infections. For example, the toxin oosporein from *Beauveria bassiana* is able to diminished cellular (e.g., reduction in the prophenoloxidase activity) and humoral (e.g., downregulation of antifungal peptides) immune responses of insects, thus contributing to the fungal replication and spread within the host hemocoels [21]. The oosporein is also able to induce dysbiosis of insect midgut microbiota and play a key role in the conversion of the asymptomatic gut symbiont to the hemocoelic pathogen, leading to the insect septicemia [22]. In addition to the infection at the level of individuals, a similar phenomenon also occurs at the level of colonies in social insect societies. Social insects in their colony are similar to cells in a body, communicate with each other and collectively work as a superorganism [23]. When fungal pathogens contaminate foragers outside the colony, they can exploit the social network to invade the colony, spread from the contaminated individuals to their naive nestmates, causing disease symptom in their group members, and finally kill the colony [1,4]. Meanwhile, fungal pathogens can also exploit the networks to spread from contaminated colonies to their neighbor colonies [1].

In this review, we mainly focus on social immunity of insects and its molecular basis in response to fungal infections. We discuss the multi-defense strategies of social immunity and illustrate how social insects exploit these defensive strategies to disrupt the process of fungal infections and deal with the fungal toxins by avoidance, resistance and tolerance at the colony level. In addition, as organization of social insect societies (i.e., social interaction networks) formed by different behavior-and-physiology members contribute to social immunity, we also discuss how these interaction networks facilitate social immunity and the molecular adaptations of the behavior-and-physiology members so as to enables us to better understand the molecular basis of social immunity. So far, studies on the defense mechanism of social immunity have been extensively reported [6,8,11,12,13,14,15,16,19,20,24]. However, the molecular basis of social immunity has yet to be fully understood such as the contribution of secreted insect venoms to social immunity and the molecular modulation of the social communication, behavior and immune priming. Thus, identifying cases of social immunity and discussing their possible molecular basis may help readers generate testable predictions for future research work in regard to the mechanism of social immunity (Figure 1).

## 2. Multi-Defense Strategies of Social Immunity

To reduce disease and improve survival of group members, social insects employ multi-defense strategies against fungal infections, which can be categorized as avoidance, resistance and tolerance [3,25] (Table 1). Specifically, avoidance is the disease defense outside insect colonies [26,27]. When the disease occurs inside insect colonies, resistance will be activated to eliminate pathogens as soon as possible [28,29] and tolerance will be employed to cover the costs of the resistance or to reduce the negative impact of the infection [3].

### 2.1. Avoidance Strategy

Avoiding infection is the first defensive line to protect insect colonies from becoming infected by preventing the entrance of pathogens into the colonies [1]. This strategy may be the best one because colonies need not suffer damage caused by infection directly [19,20,68,69] or immune response indirectly [70,71]. One important component of this strategy is to avoid direct contact with pathogens. For example, termites avoid areas containing fungal pathogens, perform vibratory warnings and close contaminated areas to prevent their nestmates from further contact with the pathogens [1,12,25,31,72]. Ants like to bring carcasses of their nestmates back for food, but they do not contact with the fungus-contaminated corpses [32]. In addition, avoidance strategy also includes the special care for materials brought into the colonies. In leaf-cutter ants, large foragers carry leaves into their colonies and hitchhikers that are a special caste of small workers on the leaves are responsible for removing fungal contaminates [33]. Similar to skin immunity of vertebrates, border defense of insect colonies is another important component of avoidance strategy. Social insects nest with antifungal materials that are collected from environments or are produced from themselves to enhance the border defense. For instance, ants collect the tree resin from environments for nesting materials to prevent fungal growth [34]. Some antifungal chemicals produced by termites, ants and bees can be also added into the materials [35,36,37,38]. Termites and ants also use symbiotic microorganism from their nesting structure to defend against fungal pathogens [39,40]. 

### 2.2. Resistance Strategy

#### 2.2.1. Sanitary Care of Contaminated Insects

Resisting infection is to eliminate pathogens quickly by clearing infectious sources and decreasing individual susceptibility in combination after insect colonies become infected by the entrance of pathogens into the colonies [1]. This defensive strategy is often accompanied by the cost of the resistance and the loss of the colony fitness [3]. Thus, the prolonged resistance is harmful to the infected colonies. To eliminate pathogens in a short time, social insects generally perform sanitary care of fungus-contaminated individuals when the pathogens initially attach loosely to the cuticles. For example, grooming behavior is an effective sanitary care to remove some disease-causing agents including fungal pathogens [4,11,15,28,29,41,73,74]. Meanwhile, social insects also disinfect the surface of the contaminated individuals by self-produced antifungal compounds to prohibit the pathogen germination and growth [17,38,44]. In termites, although soldiers are unable to groom, they provide sanitary care by producing antifungal chemicals or volatiles, contributing to the survival of the workers living together with them [42,43,75].

#### 2.2.2. Exclusion of Infected and Dead Insects

However, when fungal pathogens fully adhere to the cuticle and can no longer be cleared, they will invade body cavities of the insect and cause internal infections [19,20]. During this stage, the infected individuals and their corpses may become new infection sources inside the colonies and hence have to be excluded by themselves or their nestmates. In ants, fungus-contaminated individuals actively leave their brood chamber [45], and the contaminated foragers spent more time outside the colony and limit their area of movement inside the colony [8], and moribund individuals caused by fungal infections actively leave their nests and break off all social interactions days or hours before death [46,47]. These active self-exclusions effectively decrease the contact of naive nestmates with infectious individuals. Dying bees that may constitute a potential risk for their hives also share the active self-exclusion [24]. In addition, fungus-infected individuals and corpses inside colonies can be passively excluded by the other group members. Ants and termites behave more aggressively toward the fungus-infected workers [12,16]. Further, in termites, the infected workers are eaten when sick, and then buried after death [12,49,76]. In brood chambers of ants, workers destroy and then disinfect the infected brood [55]. In bees, workers directly remove the diseased brood from their nests [54]. As the corpses are easy to breed pathogens including fungus, undertaking behavior becomes prevalent in social insects, including removal, burial and cannibalism [77]. For instance, termites perform cannibalism by sensing an early death cue and bury dead nestmates by sensing late death cues [48]. In addition to cannibalism and corpse-burying behavior, ants also perform distant corpse removal or create ‘graveyards’ [10,32,51,52,53]. 

#### 2.2.3. Food Protection

In social insects, maintaining food quality is an effective measure to resist pathogen infections in their nests. For example, bee products such as honey, royal jelly, pollen and propolis exhibit the ability to inhibit the fungal growth [56,57]. The fungus-growing ants eliminate fungal contaminates in the garden by direct grooming and weeding, or by employing symbiont-producing chemicals as weed-killers [58,59,60]. They also create the garbage dump to place food wastes from the fungus garden and limit interactions with garbage workers [61]. These control measures efficiently stop fungal pathogens from polluting insect food and establishing in the food areas of the nests.

#### 2.2.4. Antifungal Secretions

Insect immune systems including cellular and humoral immune responses are essential for the host to resist pathogen infections [78]. In social insects, these physiological defenses are not restricted to the individual level, but also occur at the colony level. One important component of the colony-level defenses against fungal pathogens in the insect immune systems is antifungal secretions [79]—external disinfection to inhibit fungal growth—that constitute a first barrier of food [56,57], brood [17,37,38] and nest [36,37] to the pathogens and hence limit the pathogen entrance and spread. The antifungal secretions often operate in conjunction with the behavioral disease defenses described above, which contribute to the social transfer of the antifungal secretions in the fungus-infected colonies [1,2]. If the behavioral and physiological disease defenses can effectively eliminate the pathogen on the cuticles, the fungus-contaminated individuals will not face further demands for the inducible expression of immunity-related genes and consequently decrease the investment of immune costs [80,81]. Some insect venoms secreted by specific glands and shared with nestmates and nest materials such as formic acid in ants can also act as the antifungal secretions to serve colony-level protections [44], implying an important role of metabolism in social immunity, for details, see Table 2.

#### 2.2.5. Social Immunization

In addition to the external defense, insect immune systems can also serve colony-level defenses through immunological priming—enhanced internal defenses by social immunization—that confers a strong brood protection against later infection by the same fungal pathogen [82]. For instance, in termites and ants, social contact with individuals contaminated with the fungus *Metarhizium* often leads to transmission of a low pathogen dose from the contaminated individuals to their caregivers. Therefore, the caregivers contract a low-level infection that does not lead to disease symptoms, but trigger an enhanced ability to inhibit the fungal growth and hence serve nest protection against subsequent exposure to the same pathogens [14,62]. In addition, the immunological priming in social insects has some cross-generational properties, namely transgenerational immune priming (TgIP). TgIP refers to the social transfer of an acquired internal defense from the parental to the offspring generation [82,83]. For example, in the ant *Crematogaster scutellaris*, social immunization enables the queens immunized by the fungus *M. anisopliae* to enhance the antifungal activity of their offspring through TgIP, serving the brood protections [63]. If sufficiently high proportions of immune individuals through social immunization inside social insect colonies emerge, it provides ‘herd immunity’ to further limit the pathogen spread and hence protect the high-value individuals such as the queens [84] (Figure 1). 

### 2.3. Tolerance Strategy

#### 2.3.1. Nutrition and Reproduction in Tolerance

Tolerating infection is the capability of social insect colonies to cover the costs of resistance and limit negative impacts of the infection, which cannot directly eliminate the pathogens but play an important role in social immunity [3]. Although how the colonies tolerate fungal infections is unknown, we may speculate several factors that are involved in tolerance such as food, reproduction, development and metabolism. Given that insect immune systems induce a costly upregulation of immunocompetence [71,85], social insects are likely to alter their demands for the nutrition and energy. For example, a carbohydrate-rich diet contributes to social immunity in fungus-infected ants [64]. Protein nutrition benefits for shaping baseline immunocompetence and glucose oxidase activity of bees, which serves brood and nest protections, as an important component of social immunity [66]. Feeding and trophallactic behaviors are also altered by the fungus-infected ants that perform an enhanced preference to quinine and more trophallaxis with their nestmates to receive more food [65]. In addition, although workers in social insects are expendable, they are essential for maintaining their colonies and producing new queens. When the workers resist pathogen infections, both the mortality caused by the pathogens and exclusion caused by themselves or their nestmates lead to the loss of labor, which reduces the colony fitness [68,86]. To recover the fitness, social insect colonies may promote their reproductive abilities to produce more workers in case of the reduction of the worker force [67]. As only one or a few individuals in a social insect nest have reproductive capability [1] and queen reproductions may limit their immune system in female primary reproductives [87], it is dangerous for the nest that the small proportions of high-value reproductives succumb to the fungal infection [1]. Thus, the reproductives should be especially cared for, suggesting the close cooperation between resistance and tolerance strategies.

#### 2.3.2. Metabolic Tolerance

Insect metabolism is an important physiological adaptation to tolerant pathogen infections. When the pathogenic fungus *M. anisopliae* penetrates into the insect hemocoel, it produce destruxins, the cyclodepsipeptidic mycotoxins to disrupt the living cells [88]. During the interaction between the pathogen and the host, the overproduced ROS can also disrupt the living cells [89,90]. To maintain the homeostasis *in vivo*, a series of metabolic adaptations start to work, including detoxification of the fungal toxins [91] and antioxidation against the reactive oxygen species (ROS) [90]. These metabolic adaptations cannot directly eliminate pathogens but protect individuals from the infection damage, which are important parameters to quantitatively evaluate the capability of insect tolerance. For example, disease defenses in social insects are employed to prevent spreading of the pathogens from the outer nest area containing older workers (guardians and foragers) to the inner nest area containing younger nurse workers, broods and queens [92,93,94]. Given that insect pathogens in the environment are more likely to contaminate the older workers than the younger workers, broods and queens, different group members of the same age and/or caste are predicted to have distinct intrinsic tolerance capacities [3]. It would be operable to examine whether age and/or caste specific tolerance exists by comparison of their metabolic reactions in detoxification and antioxidation. Moreover, social immunization in insect colonies enables their members to not only enhance antifungal abilities, but also increase the activity of antioxidant enzymes and alter expression of proteins associated with detoxification, stress, development and other metabolism [62], suggesting an important role of metabolic tolerance in social immunization. Thus, the studies on how metabolic adaptations serve tolerance to the low-level infections and the functional mechanism of metabolic molecules in social immunity would provide a new avenue for future research regarding the metabolic regulation of social immunity.

## 3. Mechanism of Social Immunity

In insects, the mechanism of social immunity is driven by multi-level factors. Generally, the genetic elements and biochemical factors can influence individual behaviors [95,96] and physiologies [17,96], leading to changes of interactions between societal members. These altered social interactions can further affect the organization of insect colonies, which enables the colonies to limit pathogen spread and serve protections for high-value members [9]. Here, we will discuss the role of the network constituted by insect social interactions in disease defenses and give the possible molecular basis of social immunity involving in the insect communication, behavior and physiology within the insect network.

### 3.1. Social Interaction Network in Social Immunity

In social insects, colony-level disease defenses need interactions among group members, which constitutes social interaction networks to limit pathogen spread at the colony level and reduce the infection risk of individuals within the networks [6,8,97]. Generally, when individuals are contaminated with fungal pathogens, they adaptively alter their behavior [8,45,46] and physiology [20,62,98,99], and simultaneously transfer dangerous signals from the pathogen (e.g., musty odor) and themselves (e.g., volatile compounds) to the rest of naive colony members that perform colony-level disease defenses [50,54,55]. By these disease defenses, the contaminated individuals are either cared for (e.g., grooming, chemical disinfection and trophallaxis) or excluded (e.g., aggression, burial, cannibalism and removal), and their colonies are either protected against infections or abandoned [1,2,3]. This interaction network described above includes efficient communication and function through behavioral and physiological adaptations to influence disease transmission, suggesting the important role of social interaction networks in social immunity.

#### 3.1.1. The Network Structure

The social interaction network can limit pathogen spread by its structure and plasticity [8]. The network structures in insect societies are heterogeneous, leading to interaction heterogeneities that protect high-value individuals from interacting with high-risk individuals or the outside environment [1,4,6,8,9,97,100,101]. This is because members of the same age and/or caste perform similar tasks within particular compartments [4,6,99,100]. In the interaction network of generalized insect societies, queen and her brood are cared for by younger nurse workers within the compartment in the center of the colonies. In the periphery, there are many compartments and older workers within the compartments perform dangerous out-of-nest tasks such as guarding and foraging. At the edge or outside of the colonies, some older workers work within specific compartments to deal with garbage and dead bodies [1]. Usually, social interactions occur frequently within compartments rather than between compartments [9,100,101]. Especially the direct interaction between garbage workers and their nestmates is rare [61]. Thus, this heterogeneous structure of social interaction networks is an efficient barrier to prevent pathogen spread from the outer colony area to the center of colony area. In addition, compared to random networks, the structure of social interaction networks shaped by pathogen and other pressures exhibits higher modularity, lower density, larger diameter, and lower mean and maximum degree centrality, which lead to slower disease transmission, transfer of lower pathogen dose and presumably harmless pathogen load in most of individuals [8]. 

#### 3.1.2. The Network Plasticity

When detecting the presence of fungal pathogens, social insect societies rapidly share dangerous signals through social interaction networks [50,54,55] and then alter the network structures to decrease disease transmission, suggesting that the plasticity of social interaction networks contributes to disease defense in social insects [8]. The altered network structures against pathogens include strengthening of the network’s transmission-inhibiting properties (e.g. increased modularity and clustering), an increase in network distance between task groups, and decrease in the degree centrality of the contaminated individuals [8]. For example, to defend against fungal pathogens, social insects tend to gather around and groom towards the contaminated individuals [28,29,44]. Both fungus-contaminated foragers and their naive nestmates providing care increase the network distance to the rest of the colony and diminish the interactions with other naive individuals [8]. In addition, the dangerous signal from the contaminated foragers around the colony is somehow shared with nurse workers in the center of the colony and hence the nurses move the brood to increase the distance from the colony foragers [8]. When fungus-infected corpses and moribund individuals exist in the colony, their interactions within the social interaction networks are often cut off by removal, burial and cannibalism [12,32,46,49,53,54]. Similarly, the social interactions between seriously infected and naive compartments are also cut off by closing their entrance [27,30,72]. Overall, the structure of social interaction networks in the insect colonies could prevent fungal infections and is further adjusted to reinforce its ability to limit the pathogen transmission when infected, suggesting a new type of immunity known as organizational immunity (Figure 2). 

Recently, social interactions are reviewed on behavioral and physiological responses to fungus-contaminated individuals such as avoidance, resistance and tolerance [3]. Social insects are able to rapidly find the fungal pathogens and immediately adjust their behavior and physiology to form a new network structure to defend against it [8]. However, it is still unclear how social insects exchange this dangerous signal, which is an important component of social interaction against fungal and other microbial infections. As examples, dangerous signals from different producers (e.g., pathogen or host) and at different time (e.g., early cues or late cues) transferred to different task groups (e.g., older workers, younger workers, broods and queens) leads to variation in defense strategies (e.g., avoidance, resistance and tolerance). Studies on communication within social interaction networks and its role in social immunity would contribute to deeply understand the mechanism of social immunity.

### 3.2. Molecular Basis of Social Immunity

In insects, detection of and response to exogenous cues involve series of complicated behavioral and physiological adaptations that are driven by the genetic elements and biochemical factors [62,96,102], which constitutes molecular interaction networks to confer a survival advantage. For example, chemoreceptor gene family function in sensing chemical signals of odorants and tastants in the environment and drive insect behaviors through central neurons in the brain, leading to a better adaptation to the environment [103,104,105]. Pattern recognition receptors (PRRs) recognize pathogen associated molecular patterns (PAMPs) and trigger the Späetzle–Toll, immune deficiency (Imd) and/or Janus kinase–signal transducer and activator of transcription (JAK/STAT) pathways, leading to activation of insect immune systems against pathogens [78]. In social insects, disease defense is particular important due to high population densities and relatedness of individuals [1]. Consequently, social insects have evolved a rich repertoire of group behavioral defenses to quickly found the pathogen threats and stop them at the earliest moment possible [1,2,25,106], during which the behavioral defenses can even interact with and operate in conjunction with physiological defenses [44,55,79]. In combination, these defenses serve colony-level protections against multiple diseases including fungal pathogens. However, their genetic and biochemical mechanism are poorly understood. Here, we have identified a few cases and discussed their possible molecular mechanism of social immunity against fungal pathogens (Table 2).

#### 3.2.1. Chemosensory Regulation of Social Immunity

Detection of pathogens and chemical communication are necessary for performing collective defense [50,54,55,103]. As compared to solitary insects, honey bees exhibit a depauperate immune repertoire ranging from pathogen recognition to production of immune proteins [113,114]. The immunocompetence is lower in bee larvae than in adult workers [115]. In termite, reproductions are traded off against immunity and hence reduce the immune response in female primary reproductives whose death from infection is not tolerated by their colonies [87]. To cover such shortage of physiological immunity, social insects cooperate with each other and work together to decrease the susceptibility of their colonies to pathogens and provide special care for high-value individuals such as queens and larvae, which is summarized as parts of a social immune system [1]. The defense cooperation among societal members needs precise communication for a wide range of pathogen-related chemicals [50,53,55,76] and hence a repertoire of chemoreceptor genes that underlie the evolution of complex chemical communication in social insects may be expanded. Insect perceive chemical cues with three major chemoreceptor families such as the odorant (ORs), gustatory (GRs) and ionotropic (IRs) receptors [105]. Indeed, the repertoire of ORs is encoded by about 60 genes in the fly *Drosophila melanogaster* genome [116]. However, the *OR* gene family is dramatically amplified in the ants *Camponotus floridanus*, *Harpegnathos saltator* [117] and the honey bee *Apis mellifera* [118] with 352, 347 and 163 *OR* genes respectively. Similarly, the IRs are more abundant in the termite *Zootermopsis nevadensis* (150 *IR* genes) than in the *D. melanogaster* (66 *IR* genes) [109]. This rich repertoire of chemoreceptor genes implies a molecular basis of the enhanced chemosensory function in social insects, driving the social interaction networks against pathogens before pathogen causing infections.

Although little is reported about the detection and communication mechanism of social insects induced by fungal pathogens up to now, people could see that chemosensory genes should be an important component of molecular basis of social immunity. The *orco* gene encodes the obligate co-receptor of all ORs and plays a critical role in olfaction [117,119]. Recent study showed that the *orco* mutant ants *H. saltator* and *Ooceraea biroi* lose their olfactory function, leading to reduced response to odorants, disability of communication with conspecifics and disordered social behavior [95,107]. *A. mellifera* exhibit significantly altered antenna proteins during resistance against *Varroa destructor* mite, in which an odorant binding protein shows strong correlation with hygienic behavior (HB) [108]. Similarly, when resisting fungal pathogens, adult workers perceive specific volatile chemicals from the infected larvae and perform the HB [54]. In the termite *Coptotermes formosanus*, fungal odor enhances mutual grooming and attack of the contaminated individuals [50]. However, removal of the termite antennae led to disrupted grooming behavior towards fungus-contaminated individuals [120]. Together, we suggest a close relationship between olfaction and social immunity against fungal pathogens, in which *orco* and *OR* genes may function in modulating social immunity in insects. 

Chemoreception is able to facilitate social immunity by influencing not only the behavior defense, but also the physiological defense. In bees, societal members can immunize themselves by directly contacting with cuticular hydrocarbon (CHC) cues of immune-challenged workers [121,122,123], indicating a potential route to social immunization against bacterial pathogens in social insect societies. In addition, it has been reported that social transfer of low-dose fungal conidia promotes social immunization in social insects [14,62]. However, whether and how non-volatile CHC cues of fungus-infected individuals could trigger an upregulation of antifungal activity of caregivers without low-level infections need to be further studied.

#### 3.2.2. Physiological Regulation of Social Immunity

Physiological adaptations are employed at multiple levels to fight infectious diseases such as the external disease defense that enable insects to prevent infections before pathogen causes damage, and inducible immunity imposes costs [79]. For example, skin or integument immunity protects newly molted insects from airborne fungal infection using prophenoloxidases (PPOs) in molting fluids, suggesting an extended arm of insect immune systems [124]. In social insects, the physiology-mediated external defense is particularly important because social insects often inhabit in the pathogen-rich environment and some of them perform dangerous tasks out of the colonies [2,4]. To facilitate colony defense against the omnipresent pathogens, social insects such as termites, ants and bees secrete antimicrobial peptides and metabolites for external defense, which not only protects the producers themselves but also serves several altruistic purpose, such as food, brood and nest protections (Table 1). 

To verify the physiology-mediated external defense against fungal pathogens, Hamilton and Bulmer used RNAi technology to knock down the expressions of two immune genes (*Gram-negative bacteria binding protein 2* and *termicin*) in the termite *Reticulitermes flavipes*. They found that the silenced termites exhibit significantly decreased antifungal activity of the cuticle washes and increased mortality caused by fungal infections [17]. In the ant *Lasius neglectus*, the metabolite formic acid is an important antifungal component in chemical disinfection of the pupae. Tragust and his colleagues stopped the ants from using the formic acid by mouth blockage, and then found a significant reduction of the antifungal activity of the pupal surface and a significant increase of the mortality of the fungus-infected pupae [48]. Venom peptides such as Melittin are capable of antifungal activity [125]. In the honey bee *A. mellifera*, by using Matrix-assisted laser desorption/ionization tandem mass spectrometry (MALDI-TOF/TOF MS) technology, two venom peptides, namely Melittin and Apamin, were found to be smeared on the body surface of females [38]. These studies reveal the mechanism of social immunity driven by immune and metabolic molecules. Furthermore, as we know, animal molecular adaptations are not isolated and they tend to directly or indirectly interact with each other within or between pathways, suggesting a molecular interaction network *in vivo* [126]. For example, Toll-like receptors are able to recognize fungal pathogens and trigger an antifungal immune response of insects [110,111]. However, such immune receptors also function in regulation of metabolism and activation of inflammatory pathways during the pathogenesis of metabolic diseases in human beings [127]. Thus, we predict a more complex physiological adaptation driving mechanism of social immunity including the external defense against fungal pathogens. Indeed, social insects challenged with fungal pathogens exhibit a series of molecular adaptations involving in the immune signal, immune effector, detoxication, antioxidation, energy metabolism, biosynthesis, development and other unknown functions [62,99,112]. Among these molecules, transglutaminase (TG) that predictively functions in clotting against fungal pathogens in termites [62] was also identified highly correlated with reduced hive infestation by the *V. destructor* in honey bees [96]. We suggest that TG is a key molecule in regard to modulation of social immunity against multiple disease infections in both *Hymenoptera* and *lsoptera* social insects, possibly through its facilitation to ‘herd immunity’ [84]. In addition, histones were also identified significantly upregulated in termites during fungal infections [62]. Histones are well known for its precise modulation of innate immune and inflammatory responses by the modifications such as acetylation [128], methylation [129] and phosphorylation [130]. Meanwhile, histones were also reported as immune effectors that are secreted for skin immunity of humans against fungal infections [131] and for internal immunity of shrimp against bacterial infections [132]. Thus, we predict that histones may be also an important component of the molecular basis of social immunity against fungal pathogens through its facilitation to the external defense and/or herd immunity. TG, histone and other identified genes and proteins in social insects need further function verification.

Another important component of physiological adaptations in social immunity is immune priming, contributing to social immunization in insects. Recent study shows that the innate immune system of invertebrates shares some properties of the adaptive immune system of vertebrates such as immunological memory, implying a robust and specific immune response in invertebrates including insects [62,133]. This immunological memory involves introduction of an initial infection to activate innate immune responses and then confer a strong protective effect against the later infection by the same pathogen [134,135,136]. In social insects, for instance, an initial infection of societal members by the fungus *M. anisopliae* often causes the disease outbreak. When societal members are initially immunized by a low-dose fungal pathogen from the contaminated nestmates, they will exhibit a lower susceptibility to the same fungal pathogens than before, leading to herd immunity to limit the disease outbreak [14,45,62,137]. However, this immunological memory cannot serve brood protection against a different pathogen. Immunization by the fungal infection does not lead to a protective effect against bacteria [14] and the caregivers immunized by *Metarhizium* become more vulnerable to subsequent exposure to a different fungal pathogen *Beauveria* in ant colonies [16]. These future susceptibilities caused by immunization indicate the memory and specificity in the insect immune systems. In social insect immunization, whether this protective effect of immune priming is due to ‘immunological loitering’ or an enhanced ability to generate second immune responses is unknown. In social insects, protective effects of immune priming involve not only immune responses, but also behavioral defenses. In the ants *Acromyrmex echinatior* and *Formica selysi*, workers primed by the fungus *M. anisopliae* shows more frequent grooming behavior than non-primed workers [29,138]. Besides, physiological adaptations in social immunity include TgIP, influencing the disease susceptibility of the offspring generation in social insect colonies [63,82,83]. These phenomena described above highlight the multiple functions of immune priming in social immunity, serving physiological, behavioral and transgenerational protections. Thus, shedding light on the molecular mechanism of immune priming is considered worthwhile to deeply study. 

However, social insects’ immune system is not always primed successfully upon an initial infection of some fungal pathogens. For example, the ants *F. selysi* and *L. neglectus* primed by the fungus *B. bassiana* displays no survival benefit when later infection by the same pathogen [139], suggesting an obvious difference in protective effects in social insects against different fungal pathogens. This may be due to the less virulence of the fungal pathogens those cause a weaker immune reaction in ants [16] or the small repertoire of the ant PRRs that are specific to the fungal pathogens [133], of which no evidence has been presented up to now.

## 4. The Role of Fungal Toxins in the Evolution of Social Immunity

As we known, the evolution of host immune response is largely driven by pathogens that they encounter [140]. To combat different disease-causing agents such as fungus, bacteria and viruses, insects and the other animals have evolved different immune pathways [78]. The defensive responses of both solitary and social insects start with avoidance to stop direct contact with pathogens [2,3,25,141]. When solitary insects are infected, their immune systems are activated. Body cells function in the pathogen recognition, information transfer, killing pathogens by encapsulation (cellular immunity) and/or antifungal peptides (humoral immunity) and even the programmed cell death (apoptosis) [70,78,142,143]. Similarly, when the social insects’ colonies like superorganisms are infected, their ‘social immune systems’ are activated. Societal members perform the defensive functions as the body cells do, including pathogen recognition, chemical communication, killing pathogens by burial (‘social encapsulation’) and/or antifungal secretions (similar to humoral immunity) and active social exclusion (‘social apoptosis’) [1,2,3,23]. Lastly, both solitary and social insects take a special care of their high value cells or individuals, germ lines or queens, to prevent pathogen infections [23]. Recently, some researchers have reported that fungal toxins could be directly recognized by the PRRs and further activate host innate immune responses [142]. However, how ‘social immune systems’ of social insect colonies directly monitors fungal toxins and discriminates fungal toxins from bacterial toxins are unclear. Some studies regarding the interaction between ‘sickness or death cues’ and social immunity [48,54,55] implied the indirect colony-level recognition of fungal toxins. Although the mechanism of the toxin recognition is unknown, fungal toxins are indeed able to induce social immunity performed by social insect workers to maximize their inclusive fitness. For example, ants contaminated with live fungal conidia actively left the brood chamber and induced the enhanced brood care, whereas ants contaminated with the inactivated conidia without virulence did not change their behaviors [45]. Eusocial termites were able to avoid the area polluted by fungal toxins [27]. We conclude that the fungal toxin may be an important driving force for the evolution of social immunity in insects against fungal pathogens.

## 5. Conclusions and Future Work

In conclusion, social insects have evolved highly complex social interactions of ‘host–fungal pathogen’ and ‘host–host’, which consists of recognition, communication and a combination of multi-defense strategies. These social insect interactions serve the colony-level protection through an interaction network, whose structure can prevent the pathogen infection and plasticity can stop the pathogen transmission. Social insects are able to rapidly detect the fungus cues and immediately perform multiple defenses depending on behavioral and physiological adaptations. While avoidance and resistance against a fungal pathogen has been reported, little is currently known about how social insects perceive pathogen information and how colony-level tolerance facilitates social immunity.

Additionally, we predicted that the chemosensory and physiological mechanism of social insects may play a critical role in driving social immunity against fungal pathogens, which needs to be further studied. As a likely example, chemoreceptors such as ORs may recognize fungal pathogens and induce behavioral disease defenses. In social insects, the behavioral disease defenses often cooperate with physiological disease defenses to provide protections at the colony level [1,2]. The physiology-mediated antifungal secretions [79] and immune priming [82] are used for external and internal disease defense respectively. These combined defenses modulated by different molecules reveal a molecular basis of social immunity, of which more evidences need to be presented.

In the future work, due to a well-established framework, an increasing knowledge, and more tools involved in studies of social immunity [3], it is feasible to experimentally address the questions in regard to social immunity and its molecular mechanism. For example, an automated ant tracking system has been used for organization immunity [8]. The proteomics technology and correlation analysis have been reported for genetic and biochemical mechanism of social immunity [96,108]. RNAi, confocal laser scanning microscopy (CLSM) and gene editing have been reported to reveal the genetic and epigenetic regulation of social behavior [95,107,144]. Thus, we can select some of these methods and novel methods in combination to enrich the theory of social immunity.

## Figures and Tables

**Figure 1 toxins-11-00244-f001:**
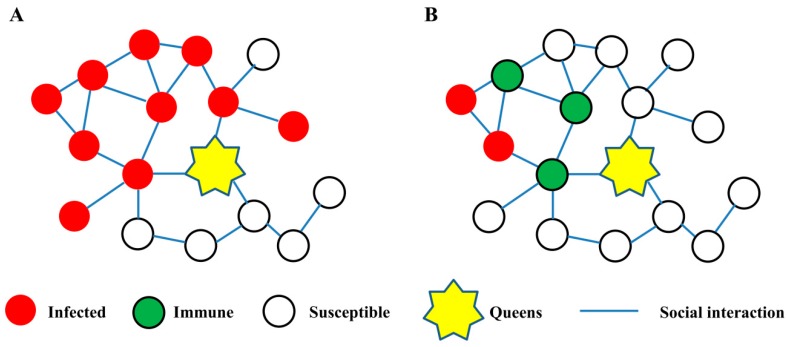
‘Herd immunity’ in social insect colonies. (**A**) Fungal pathogens are expected to employ social interaction networks to spread from the infected to susceptible individuals and risk infecting the queens inside the colonies. (**B**) When high numbers of immune individuals through immunization exist inside the colonies, they will form an immune wall to cut off the spreading pathway and serve a colony-level protection.

**Figure 2 toxins-11-00244-f002:**
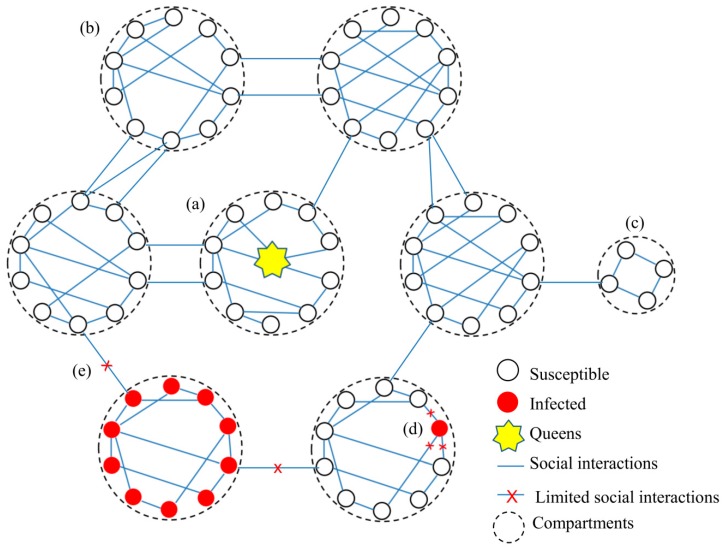
Organizational immunity in a generalized social insect colony. (**a**) The compartments consisting of queens and her broods are specially cared in the central area; (**b**) the compartments consisting of older workers serve protections in the periphery to prevent environmental pathogens from entrance into the central area; (**c**) the compartments consisting garbage and dead bodies are far away from and are stopped from direct interaction with the central area to protect queens from pollutions. In addition, the social interaction occurs more frequently within compartments than between compartments. When (**d**) individuals were infected, the social interactions between the infected and naïve individuals were limited within compartments. However, when (**e**) compartments were infected, the rest of the compartments would close the entrance and hence cutoff the interaction between the infected and naïve compartments. These managements are effective to limit the pathogens transmission and protect high-value individuals in social insect colonies.

**Table 1 toxins-11-00244-t001:** Multi-defense strategies against fungal pathogens in social insects.

Strategy	Effect	Defense Mechanism	Host	Species and Reference
Avoiding infection	Protect insect colonies from becoming infected by preventing the entrance of pathogens into the colonies	Avoid fungus-infected areas	Termites	*Macrotermes michaelseni* [27]
			Ants	*Acromyrmex striatus* [30]
		Avoid fungus-infected individuals	Termites	*Zootermopsis angusticollis* [31]
				*Reticulitermes flavipes* [12]
			Ants	*Formica rufa* [32]
		Check before colony entrance	Ants	*Atta sexdens* [33] *Atta laevigata* [33]
		Collect environmental compounds for nest materials	Ants	*Formica paralugubris* [34]
		Use self-produced compounds for nest materials (antifungal secretions)	Termites	*Zootermopsis angusticollis* [35]
				*Nasutitermes corniger* [36]
			Ants	*Acromyrmex subterraneus* [37]
				*Polyrhachis dives* [37]
			Bees	*Apis mellifera* [38]
		Use symbiotic microorganism for nest materials	Termites	*Coptotermes formosanus* [39]
			Ants	*Acromyrmex octospinosus* [40]
Resisting infection	Eliminate pathogens quickly by clearing infectious sources and decreasing individual susceptibility in combination	Grooming	Termites	*Zootermopsis angusticollis* [11]
				*Coptotermes formosanus* [28]
			Ants	*Acromyrmex echinatior* [29]
				*Solenopsis invicta* [15]
			Bees	Unkown [41]
		Chemical disinfection (antifungal secretions)	Termites	*Reticulitermes flavipes* [17]
				*Nasutitermes costalis* [42]
				*Nasutitermes nigriceps* [42]
				*Reticulitermes speratus* [43]
			Ants	*Acromyrmex subterraneus* [37]
				*Polyrhachis dives* [37]
				*Lasius neglectus* [44]
			Bees	*Apis mellifera* [38]
		Active self-exclusions	Ants	*Lasius neglectus* [45]
				*Lasius niger* [8]
				*Temnothorax unifasciatus* [46]
				*Myrmica rubra* [47]
			Bees	*Apis mellifera* [24]
		Aggressive behavior	Termites	*Reticulitermes flavipes* [12]
			Ants	*Lasius neglectus* [16]
		Cannibalism/Burial	Termites	*Reticulitermes flavipes* [12,48]
				*Coptotermes formosanus* [49,50]
			Ants	*Formica rufa* [32]
				*Temnothorax lichtensteini* [51]
		Removal	Ants	*Myrmica rubra* [52]
				*Solenopsis invicta* [53]
			Bees	*Apis mellifera* [54]
		Destructive disinfection	Ants	*Lasius neglectus* [55]
		‘Graveyards’	Ants	*Solenopsis invicta* [10]
		Food protection (antifungal secretions)	Bees	*Apis mellifera* [56,57]
			Ants	*Atta colombica* [58]
				*Acromyrmex species* [59]
				*Tribe Attini* [60]
		‘Garbage dump’	Ants	*Atta cephalotes* [61]
		Social immunization	Termites	*Reticulitermes chinensis* [62]
			Ants	*Lasius neglectus* [14]
				*Crematogaster scutellaris* [63]
Tolerating infection	Cannot directly eliminate pathogens but play an important role in social immunity	Food/nutrition	Ants	*Ectatomma ruidum* [64]
				*Solenopsis invicta* [65]
			Bees	*Apis mellifera* [66]
		Reproduction	Ants	*Cardiocondyla obscurior* [67]
		Detoxification/antioxidation	Termites	*Reticulitermes chinensis* [62]

**Table 2 toxins-11-00244-t002:** Molecular basis of social immunity against fungal pathogens in social insects.

Regulator	Function	Origin	Molecule	Species and Reference
Chemosensory regulation	Detecting pathogens, chemical communication and inducing behavioral and physiological defenses	Fungal pathogens	Odor substances	Termites: *Macrotermes Michaelseni* [27] *Coptotermes formosanus* [50] *Reticulitermes flavipes* [48]
		Host	Chemical ‘sickness cues’	Ants: *Lasius neglectus* [55]
			Linoleic and oleic acids	Ants: *Solenopsis invicta* [53]
			Phenethyl acetate	Bees: *Apis mellifera* [54]
			*OR* and *Orco* genes	Ants: *Ooceraea biroi* [95] *Harpegnathos saltator* [107] Bees: *Apis mellifera* [108]
			*IR* genes	Termites: *Zootermopsis nevadensis* [109]
Physiological regulation	External defense by sharing insect venoms with their nestmates and nest materials	Frontal gland	α-pinene and limonene	Termites: *Nasutitermes costalis* and *N. nigriceps* [42]
			(-)-β-elemene	Termites: *Reticulitermes speratus* [43]
		Oral secretions	Proteins and chemicals	Termites: *Mastotermes darwiniensis* [75]
		Fecal material	Unknown	Termites: *Zootermopsis angusticollis* [35]
		Salivary gland	Termicins and GNBPs	Termites: *Reticulitermes flavipes* [17] *Nasutitermes corniger* [36]
		Venom gland	Formic acid	Ants: *Lasius neglectus* [44]
			Melittin	Bees: *Apis mellifera* [38]
		Metapleural gland	Unknown	Ants: *Lasius neglectus* [14] *Acromyrmex subterraneus* [37]
		Hypopharyngeal gland	Royal jelly	*Apis mellifera* [56,57]
	Internal defense by enhancing physiological resistance and tolerance to fungal infections	Immune signal and immune effector	Toll pathway	Model insect: *Drosophila* [110,111]
			Termites: transglutaminase and histone Ants: β-1,3-glucan-binding protein and defensin Bees: tyrosine kinase 3, MyD88 and abaecin	Termites: *Reticulitermes flavipes* [17] *Reticulitermes chinensis* [62] Ants: *Lasius neglectus* [14] *Acromyrmex echinatior* [112] Bees: *Apis mellifera* [38,99]
		Detoxication	Glutathione S-transferase and cytochrome P450	Termites: *Reticulitermes chinensis* [62]
		Antioxidation	Termites: superoxide dismutase and catalase Bees: hexamerin 70b and vitellogenin	Termites: *Reticulitermes chinensis* [62] Bees: *Apis mellifera* [99]
		Energy metabolism, biosynthesis, development and others	Others	Termites: *Reticulitermes chinensis* [62] Ants: *Acromyrmex echinatior* [112] Bees: *Apis mellifera* [99]
Other	Nest materials	Conifer (*Picea abies*)	Resin	Ants: *Formica paralugubris* [34]
		*Streptomyces*	Unknown	Termites: *Coptotermes formosanus* [39]
		*Streptomyces*	Candicidin and antimycins	Ants: *Acromyrmex octospinosus* [40]

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
