# Peer review of "The Mechanisms of Social Immunity Against Fungal Infections in Eusocial Insects"

_toxins, 2019, doi:10.3390/toxins11050244_

Reviewer 1 Report

The present review summarizes behavioral and molecular mechanisms potentially involved in insect social immunity against fungal pathogens. Although it covers most of the recent advances in the research field of social insects, there are some issues to be addressed to increase its impact, as listed below.

[Comment #1]

As clearly stated in the “Aims” of the journal website (https://www.mdpi.com/journal/toxins/about), this journal covers toxinology and biotoxins from any living source. Based on this aim, I felt odd that only small portion of this review mentioned something about fungal toxins (lines 195-200; ~1% of the entire manuscript length). Although it is understandable that not many studies have unveiled the relationship between fungal toxins and social immunity at molecular level, authors should make effort to discuss more about the potential effects of toxins on insect social behaviors. Some fungal toxins are known to affect insect immunity and physiology, as recently reported by Feng P, et al. 2015 (https://www.ncbi.nlm.nih.gov/pubmed/26305932) and Wei G, et al. 2017 (https://www.ncbi.nlm.nih.gov/pubmed/28533370), which could be potentially discussed. Also, just because not much is known about fungal toxins themselves, authors could even raise the application of non-social insect models to experimentally study the basic toxicity of fungal toxins to motivate the readers, as recently covered by Matsumoto Y, et al. 2019 (https://www.ncbi.nlm.nih.gov/pubmed/30666711).

Without these efforts to discuss about toxins, I feel this review is much suited for other journals that focus on insect behaviors and immunology, but not toxins.

[Comment #2]

Overall the manuscript lacks discussions from the viewpoint of evolution. The authors only mentioned that “… honey bees exhibit a depauperate immune repertoire ranging from pathogen recognition to production of immune proteins” (lines 307-308) by citing two references, which I feel not enough. The motivation to develop social immunity in insects should have been driven largely by pathogens that insects encountered in the course of evolution, analogous to physiological changes accompanied by bacterial symbionts. What is the possible driving force for the evolution of social immunity in insects against fungal pathogens (especially focusing on fungal toxins)?

In addition, the authors did not provide sufficient information to discriminate immunity against bacteria or fungi at molecular level, in section 3.2. It is well known in Drosophila melanogaster that distinct immune pathways are activated by different PAMPs of pathogens; fungal glucans recognized by GNBP3 and bacterial Lys-type peptidoglycans bound to PGRP-SA commonly stimulate the Spaeztle-Toll-Dorsal/Dif pathway, whereas DAP-type peptidoglycans from gram-negative bacteria bind to long-form PGRPs followed by the Imd-Relish pathway. Importantly, Gottar M et al. have reported that a fungal virulence factor PR1 can activate the Toll pathway via proteolytic cleavage of Drosophila protease named Persephone (https://www.ncbi.nlm.nih.gov/pmc/articles/PMC1865096/). Based on these literatures, authors may discuss in more detail about the differences in social immunity against bacterial versus fungal toxins, from an evolutionary standpoint.

[Comment #3]

The authors claimed that they will “mainly focus on social immunity of insects and its molecular basis in response to fungal pathogens” (lines 49-50). Based on their statement, I expected to see figures/tables summarizing both “social immunity” and “its molecular basis” in similar depth. However, authors provided a detailed table only for the former, but not for the latter. As the section 3.2 is rather lengthy and hard to read, I would recommend summarizing the molecular factors that potentially affect social immunity mentioned in this section as a concise figure/table, as done in Table 1.

Related to this, there are some redundancies within the manuscript that make it lengthy. Information regarding social immunization by fungal infection and transgenerational immune priming (TgIP) in section 2.2.5 overlaps with some parts of section 3.2.2 (lines 396-416). Authors should omit either one or discuss them in different ways.

[Comment #4]

I would suggest omitting Figure 1, which is difficult to follow. Why are there several arrows shown in different size and color? Why is “Resistance/Tolerance/Avoidance” shown in Venn diagrams; what do the overlaps mean? How could “Resistance/Tolerance/Avoidance” be affected by both “Organizational adaptations” and “Behavioral adaptations”? This figure raises these confusions and does not contain valuable information.

[Comment #5]

Authors need to pay more attention to terminology. They use technical terms closely related to “social immunity” without adequate explanation, such as “organizational immunity” (line 276) and “herd immunity” (line 387). Since “social immunity” is the main topic of this article, I would suggest clarifying the differences/meanings of these terms.

 [Comment #6]

I would strongly suggest asking for professional English editing. Singular and plural usages are mixed up (e.g., “In facts, …” in line 30), and some sentences have complicated structures that make them hard to read (e.g., lines 252-256).

Author Response

Reviewer 1

Comments and Suggestions for Authors

The present review summarizes behavioral and molecular mechanisms potentially involved in insect social immunity against fungal pathogens. Although it covers most of the recent advances in the research field of social insects, there are some issues to be addressed to increase its impact, as listed below.

Comment 1: As clearly stated in the “Aims” of the journal website (https://www.mdpi.com/journal/toxins/about), this journal covers toxinology and biotoxins from any living source. Based on this aim, I felt odd that only small portion of this review mentioned something about fungal toxins (lines 195-200; ~1% of the entire manuscript length). Although it is understandable that not many studies have unveiled the relationship between fungal toxins and social immunity at molecular level, authors should make effort to discuss more about the potential effects of toxins on insect social behaviors. Some fungal toxins are known to affect insect immunity and physiology, as recently reported by Feng P, et al. 2015 (https://www.ncbi.nlm.nih.gov/pubmed/26305932) and Wei G, et al. 2017 (https://www.ncbi.nlm.nih.gov/pubmed/28533370), which could be potentially discussed. Also, just because not much is known about fungal toxins themselves, authors could even raise the application of non-social insect models to experimentally study the basic toxicity of fungal toxins to motivate the readers, as recently covered by Matsumoto Y, et al. 2019 (https://www.ncbi.nlm.nih.gov/pubmed/30666711). Without these efforts to discuss about toxins, I feel this review is much suited for other journals that focus on insect behaviors and immunology, but not toxins.

Response 1: Our manuscript was written for the special issue of Toxins—“Fungal Infestations in Humans, Animals, Crops”—whose aims and scope were to present a collection of studies on different aspects of fungal infections: case studies, mechanisms, diagnosis, and prevention (https://www.mdpi.com/journal/toxins/special_issues/fungal_infestation_human_animal_crop). In this review, we mainly discussed how social insect employed behavioral and physiological mechanism to prevent fungal infections at the colony level, which is involved in insect social immunity. For example, the avoidance and external resistance may be the most directly effective preventions to stop fungus poisoning during infections among multi-level preventions in social insect colonies. When fungal pathogens succeed in establishment and spread inside the insect colonies, the chemical disinfection is also an important prevents to protect societal members from the fungus poisoning. Some of disinfection agents are insect venoms secreted by the specific poison and other glands, and the other disinfection agents are antiseptics from the environments and symbiotic microorganisms. Therefore, our manuscript is suited for Toxins. Additionally, we have added some new contents to discuss fungal toxins (Line 45-52; Line 493-518) and insect venoms (line 168-171) in the revised version.

Comment 2: Overall the manuscript lacks discussions from the viewpoint of evolution. The authors only mentioned that “… honey bees exhibit a depauperate immune repertoire ranging from pathogen recognition to production of immune proteins” (lines 307-308) by citing two references, which I feel not enough. The motivation to develop social immunity in insects should have been driven largely by pathogens that insects encountered in the course of evolution, analogous to physiological changes accompanied by bacterial symbionts. What is the possible driving force for the evolution of social immunity in insects against fungal pathogens (especially focusing on fungal toxins)?

In addition, the authors did not provide sufficient information to discriminate immunity against bacteria or fungi at molecular level, in section 3.2. It is well known in Drosophila melanogaster that distinct immune pathways are activated by different PAMPs of pathogens; fungal glucans recognized by GNBP3 and bacterial Lys-type peptidoglycans bound to PGRP-SA commonly stimulate the Spaeztle-Toll-Dorsal/Dif pathway, whereas DAP-type peptidoglycans from gram-negative bacteria bind to long-form PGRPs followed by the Imd-Relish pathway. Importantly, Gottar M et al. have reported that a fungal virulence factor PR1 can activate the Toll pathway via proteolytic cleavage of Drosophila protease named Persephone (https://www.ncbi.nlm.nih.gov/pmc/articles/PMC1865096/). Based on these literatures, authors may discuss in more detail about the differences in social immunity against bacterial versus fungal toxins, from an evolutionary standpoint.

Response 2: Thank you so much for your valuable suggestions. In this revision, we discussed detailedly the role of fungal toxins in the evolution of social immunity (Line 493-518).

Comment 3: The authors claimed that they will “mainly focus on social immunity of insects and its molecular basis in response to fungal pathogens” (lines 49-50). Based on their statement, I expected to see figures/tables summarizing both “social immunity” and “its molecular basis” in similar depth. However, authors provided a detailed table only for the former, but not for the latter. As the section 3.2 is rather lengthy and hard to read, I would recommend summarizing the molecular factors that potentially affect social immunity mentioned in this section as a concise figure/table, as done in Table 1.

Related to this, there are some redundancies within the manuscript that make it lengthy. Information regarding social immunization by fungal infection and transgenerational immune priming (TgIP) in section 2.2.5 overlaps with some parts of section 3.2.2 (lines 396-416). Authors should omit either one or discuss them in different ways.

Response 3: Thanks for your valuable recommendation. In this new version, we have added the Table 2 titled “Molecular basis of social immunity against fungal pathogens in social insects”, including the molecular factors and their potential functions in social immunity. In section 2.2.5, we rewrite the overlaps and mainly discuss the social immunization in social insect societies. Additionally, in section 3.2.2, we rewrite the overlaps and mainly discuss the immune priming in social insects.

Comment 4: I would suggest omitting Figure 1, which is difficult to follow. Why are there several arrows shown in different size and color? Why is “Resistance/Tolerance/Avoidance” shown in Venn diagrams; what do the overlaps mean? How could “Resistance/Tolerance/Avoidance” be affected by both “Organizational adaptations” and “Behavioral adaptations”? This figure raises these confusions and does not contain valuable information.

Response 4: Thanks for your suggestion, Figure 1 has been omitted in this revision.

Comment 5: Authors need to pay more attention to terminology. They use technical terms closely related to “social immunity” without adequate explanation, such as “organizational immunity” (line 276) and “herd immunity” (line 387). Since “social immunity” is the main topic of this article, I would suggest clarifying the differences/meanings of these terms.

Response 5: Thank you so much for your valuable suggestions. We have added some new Figures (Figure 1 and 2) to further explain the ‘Herd immunity’ and ‘organizational immunity’, both of which belong to ‘social immunity’.

Comment 6: I would strongly suggest asking for professional English editing. Singular and plural usages are mixed up (e.g., “In facts, …” in line 30), and some sentences have complicated structures that make them hard to read (e.g., lines 252-256).

Response 6: Thanks for your suggestion and the language of our manuscript in this revised version has been edited by a native English speaking colleague.

Reviewer 2 Report

The authors presented the review on the mechanisms of social immunity against fungal pathogens in eusocial insect. The content of this manuscript is well organized. This manuscript contains content that is of interest to experts in this field as well as non-experts. If possible, the authors should add diagrams that illustrate the individual strategies and mechanisms of social immunity and compounds that have activity against fungal pathogens to help readers of this journal.

Author Response

Reviewer 2

Comments and Suggestions for Authors

The authors presented the review on the mechanisms of social immunity against fungal pathogens in eusocial insect. The content of this manuscript is well organized. This manuscript contains content that is of interest to experts in this field as well as non-experts. If possible, the authors should add diagrams that illustrate the individual strategies and mechanisms of social immunity and compounds that have activity against fungal pathogens to help readers of this journal.

Response: Thanks for your affirmation and suggestion. We have added the Table 2 titled “Molecular basis of social immunity against fungal pathogens in social insects”, including individual adaptations and their function in social immunity and some antifungal compounds.

Reviewer 3 Report

Dear Authors,

In this review: toxins-484485-v1 entitled “Advances in the mechanisms of social immunity against fungal pathogens in eusocial insects”, the Authors analyzed a series of unique disease defenses at the colony level, which consists of behavioral and physiological adaptations of some insects. The article contains a lot of interesting data and helps to understand some of the mechanism of social immunity in eusocial insects such as termites, ants and bees. Manuscript is well written, organized and valuable.

I think, however, that article is not enough thematically linked with the journal “Toxins” and the subject of this manuscript is more appropriate for the journal “Insects” than for “Toxins”. Therefore, I suggest Authors to send an article there.

With highest regards,

Author Response

Reviewer 3:

Comments and Suggestions for Authors

Dear Authors,

In this review: toxins-484485-v1 entitled “Advances in the mechanisms of social immunity against fungal pathogens in eusocial insects”, the Authors analyzed a series of unique disease defenses at the colony level, which consists of behavioral and physiological adaptations of some insects. The article contains a lot of interesting data and helps to understand some of the mechanism of social immunity in eusocial insects such as termites, ants and bees. Manuscript is well written, organized and valuable.

I think, however, that article is not enough thematically linked with the journal “Toxins” and the subject of this manuscript is more appropriate for the journal “Insects” than for “Toxins”. Therefore, I suggest Authors to send an article there.

With highest regards,

Submission Date

29 March 2019

Date of this review

14 Apr 2019 23:28:30

Response:

Our manuscript was written for the special issue of Toxins-“Fungal Infestations in Humans, Animals, Crops”—whose aims and scope were to present a collection of studies on different aspects of fungal infections: case studies, mechanisms, diagnosis, and prevention (https://www.mdpi.com/journal/toxins/special_issues/fungal_infestation_human_animal_crop). In this review, we mainly discussed how social insect employed behavioral and physiological mechanism to prevent fungal infections at the colony level, which is involved in insect social immunity. For example, the avoidance and external resistance may be the most directly effective preventions to stop fungus poisoning during infections among multi-level preventions in social insect colonies. When fungal pathogens succeed in establishment and spread inside the insect colonies, the chemical disinfection is also an important prevents to protect societal members from the fungus poisoning. Some of disinfection agents are insect venoms secreted by the specific poison and other glands, and the other disinfection agents are antiseptics from the environments and symbiotic microorganisms. Therefore, our manuscript is suited for Toxins. In addition, we have added some new contents to discuss fungal toxins (Line 45-52; Line 493-518) and insect venoms (line 168-171) in this revision.

Round  2

Reviewer 3 Report

Dear Authors,

I've reviewed the guidelines for the special issue of Toxins: “Fungal Infestations in Humans, Animals, Crops” and indeed this issue will present a collection of studies on different aspects of fungal infections: case studies, mechanisms, diagnosis, and prevention.

In this revised review: toxins-484485-revised 1 entitled “The mechanisms of social immunity against fungal infections in eusocial insects”, the Authors have slightly modified the manuscript and have added some new contents about fungal toxins or fungal infections and their role in the evolution of social insects immunity.

Therefore, I accept this article in present form.

With highest regards,